# New Viruses Infecting Hyperthermophilic Bacterium *Thermus thermophilus*

**DOI:** 10.3390/v16091410

**Published:** 2024-09-03

**Authors:** Matvey Kolesnik, Constantine Pavlov, Alina Demkina, Aleksei Samolygo, Karyna Karneyeva, Anna Trofimova, Olga S. Sokolova, Andrei V. Moiseenko, Maria Kirsanova, Konstantin Severinov

**Affiliations:** 1Institute of Gene Biology Russian Academy of Sciences, 119334 Moscow, Russia; matveykolesnik@gmail.com (M.K.); annat@genebiology.ru (A.T.);; 2Faculty of Biology, MSU-BIT University, Shenzhen 518172, China; 3Faculty of Biology, Lomonosov Moscow State University, 119234 Moscow, Russia; 4Waksman Institute, Rutgers, The State University of New Jersey, Piscataway, NJ 08854, USA

**Keywords:** bacteriophage, CRISPR-Cas system, mobile genetic elements

## Abstract

Highly diverse phages infecting thermophilic bacteria of the *Thermus* genus have been isolated over the years from hot springs around the world. Many of these phages are unique, rely on highly unusual developmental strategies, and encode novel enzymes. The variety of *Thermus* phages is clearly undersampled, as evidenced, for example, by a paucity of phage-matching spacers in *Thermus* CRISPR arrays. Using water samples collected from hot springs in the Kunashir Island from the Kuril archipelago and from the Tsaishi and Nokalakevi districts in the Republic of Georgia, we isolated several distinct phages infecting laboratory strains of *Thermus thermophilus*. Genomic sequence analysis of 11 phages revealed both close relatives of previously described *Thermus* phages isolated from geographically distant sites, as well as phages with very limited similarity to earlier isolates. Comparative analysis allowed us to predict several accessory phage genes whose products may be involved in host defense/interviral warfare, including a putative Type V CRISPR-*cas* system.

## 1. Introduction

Terrestrial hot springs are inhabited by diverse microorganisms capable of thriving at high temperatures. Thermophilic bacteria of the *Thermus* genus are widespread in hot springs with temperatures above 60 °C and neutral pH, in self-heating compost piles, as well as in industrial water heating systems [1]. Historically, representatives of the genus have been an important source of thermostable enzymes, most notably, the Taq (*Thermus aquaticus*) DNA polymerase widely used for PCR. In addition, *Thermus* proteins have been a popular object of crystallographic studies because of their robust structures [2].

A number of immunity mechanisms of *Thermus*, including multiple CRISPR-Cas [3] and restriction–-modification [2] systems, and prokaryotic argonaute proteins [4] have been described. The abundance of these systems implies that there exist multiple phages and mobile genetic elements that use *Thermus* as a host. Indeed, several highly diverse *Thermus* phages have been isolated over the years. Many of these phages are unique, rely on highly unusual developmental strategies, and encode novel enzymes [5]. The variety of *Thermus* phages is clearly undersampled [6]. Additional searches for *Thermus* phages, particularly at sites which have not been sampled before, will likely result in novel/highly divergent viruses whose studies could benefit both basic science and biotechnology. Analysis of relatives of already known *Thermus* phages can also be highly informative. In the genomes of closely related bacteriophages, the conserved gene order is sometimes disrupted by highly variable regions encoding the so-called “moron genes” [7]. Numerous novel defense systems have been discovered in such regions [8]. Presumably, these defense systems contribute to local adaptations of phages to host strains and/or competing viruses. Studies of such adaptations can be particularly interesting in the case of *Thermus* and other thermophiles whose habitats are highly fragmented and often separated from each other by very large distances. 

In this work, we isolated several *Thermus* bacteriophages from hot springs in the Kunashir Island, Russia and from the South Caucasus mountains in the Republic of Georgia. Kunashir is a volcanic island belonging to the Great Kuril Chain. It is formed by four active volcanoes and has a large geothermal system including hot streams, lakes, and ponds with water temperatures ranging from 30 to 100 °C and pH varying from highly acidic (~1) to neutral (~8) [9]. The microbial communities of Kunashir hot springs were investigated previously using metabarcoding and metagenomic approaches and *Thermus* sequences were found in several hot springs with neutral water [10,11,12]. Yet, these studies did not assess viral diversity and, to the best of our knowledge, no phages infecting *Thermus* have been isolated on Kunashir. Another region enriched with geothermal springs is South Caucasus with a number of >60 °C hot springs located in the Republic of Georgia [13]. The microbiomes of Georgian hot springs were studied only using the metabarcoding approach and abundant *Thermus* sequences were detected [14]. Yet, no *Thermus* phages from this area are known. Using water samples collected from hot springs in Kunashir and from the Tsaishi and Nokalakevi districts in Georgia during expeditions organized in 2023, we isolated several distinct phages infecting laboratory *Thermus thermophilus* strains. Genomic sequence analysis of 11 new phages revealed both close relatives of previously described *Thermus* viruses isolated from geographically distant sites, as well as phages that could not be taxonomically placed using tools such as *tax_myPHAGE* (Appendix A). Comparative analysis allowed us to predict several accessory phage genes whose products may be involved in host defense/interviral warfare, including a putative Type V CRISPR-*cas* system that may be involved in competition with *Thermus* mobile genetic elements.

## 2. Material and Methods

### 2.1. Phage Isolation

Environmental samples were placed into sterile 50 mL tubes and stored at room temperature for 3–4 days until arrival at the laboratory, where they were further stored at +4 °C. Cultures of *T. thermophilus* HB8 or HB27 strains were grown in an orbital shaker (180 rpm) at 70 °C in *Thermus* broth (TB) (8 g/L tryptone, 2 g/L yeast extract, 4 g/L NaCl, 1 mM MgCl_2_, and 1 mM CaCl_2_ dissolved in ‘Vittel’ mineral water. Sterile MgCl_2_ and CaCl_2_ stock solutions were added to the autoclaved medium. For cultivation on solid medium, 2% agarized TB plates were used, accompanied with 0.4% top agar when necessary. Petri dishes were incubated overnight at 65 °C.

A total of 5 mL of TB medium were inoculated with a 100 μL aliquot of overnight culture of one of the *Thermus* strains and growth proceeded until OD_600_ reached ~0.4. In total, 0.2–0.5 mL of environmental sample was added and incubation was continued overnight at 70 °C with vigorous agitation. To isolate individual phage plaques, 1 mL of enrichment culture was centrifuged at 15,000× *g* for 15 min at 4 °C, and 100 μL aliquots of supernatant were added to 15 mL of melted soft (0.4%) TB agar supplemented with 300 μL of freshly grown *T. thermophilus* HB8 or HB27 cultures (OD_600_~0.4). Mixtures were shaken to ensure homogeneity, poured over 2% TB agar plates, and incubated overnight at 65 °C. Individual plaques were picked with pipette tips and purified by several passages on the host *Thermus* strain.

For phage propagation, host cultures grown until OD_600_~0.2 were inoculated with a single phage plaque and incubated for 3 h. Cultures were centrifuged at 15,000× *g* for 30 min at 4 °C and supernatants were stored at 4 °C for further use. Phage titers obtained lysates that were measured by serial 10-fold dilutions drop tests on host-inoculated top agar. Lysates with titers less than 10^7^ plaque forming units per mL were precipitated by overnight incubation on ice with 10% PEG-8000 and 1M NaCl with further centrifugation at 15,000× *g* for 30 min at 4 °C. The pellets were resuspended in 500 μL of SM buffer (50 mM Tris*HCl pH 7.5, 100 mM NaCl, 8 mM MgSO_4_).

### 2.2. DNA Extraction

In total, 450 μL of phage lysates or concentrates were supplied with 1 μL of solutions of DNase I and RNase A (10 mg/mL) and incubated at 30 °C for 1 h. Next, 50 μL of buffer (5% SDS, 100 Tris, 10 mM EDTA, pH 8.0) and 20 μL of proteinase K (20 mg/mL) were added followed by incubation at 56 °C for 1 h. The DNA was extracted using phenol-chloroform extraction technique, precipitated with ethanol, and dissolved in nuclease-free water.

Phage DNA size and integrity were evaluated by electrophoretic separation in 1% agarose gel. Samples of genomic DNA of different sizes were digested with various Type II restriction enzymes and those showing clearly distinct digestion patterns were used for sequencing.

Genomic DNA of inoviruses was extracted from infected cells after 3 h of culture inoculation with a phage plaque. Overnight cultures were diluted 1:50 and grown until OD_600_ ~0.4 when the separate plaques were added. The DNA was extracted with Thermo Scientific GeneJET Plasmid Miniprep Kit following the standard protocol. DNA concentrations were measured by Qubit 1X dsDNA High Sensitivity (HS) assay kit, and DNA purity was evaluated by NanoDrop 2000C spectrophotometer (Thermo Fisher Scientific, Waltham, MD, USA).

### 2.3. NGS Library Preparation and Sequencing

Libraries were constructed from 1 μg of genomic DNA using MGI Easy PCR-Free Library Prep Set in accordance with User Manual v1.1 (MGI Tech Co., Shenzhen, China) and cleaned up with the provided DNA Clean Beads. Quality control was performed using Qubit ssDNA Assay Kit (Thermo Fisher Scientific, Waltham, MD, USA) and 4150 TapeStation System (Agilent Technologies Inc., Santa Clara, CA, USA). Barcoded libraries were sequenced using DNBSEQ-G400 (MGI Tech Co., Ltd., Shenzhen, China) in 2 × 150 bp PE mode.

### 2.4. Computational Analysis of Viral Sequences

Read quality was assessed with FastQC v0.12.1 [15]. Adapter removing and quality trimming was performed with fastp v0.23.4 [16]. Genomes were assembled with SPAdes v3.15.3 embedded in the shovill v1.1.0 pipeline [17]. Phage genome annotation was performed with pharokka v1.4.0 [18] with further manual curation. Protein sequences were searched with hhpred and InterProScan against PDB_mmCIF70_8_Mar and InterPro databases [19,20]. To detect IN93-like prophages integrated into *Thermus* genomes, homologs of conserved IN93 major capsid proteins (MCPs) were searched in the NCBI nr protein database. Next, 20 kbp genomic regions flanking genes IN93 MCPs homologs were reannotated using the pharokka tool [18]. To determine the borders of integrated prophages, 200 bp nucleotide sequences encoded downstream of predicted transposase gene (expected to contain the attachment site) were searched with BLASTN over the entire length of the predicted prophage region to detect target duplication sites flanking the prophage. 

To assign bacteriophage genomes to taxonomy groups, the tax_*myPHAGE* pipeline was used [21].

To build graphical alignments, protein sequences of bacteriophages were aligned with the blastn tool with e-value cutoff 1 × 10^−6^; the alignments with annotations were rendered with the pygenomeviz package [22]. 

### 2.5. Electron Microscopy

The samples for the transmission electron microscopy were prepared with a negative staining technique. In total, 3 μL of samples were applied to the glow-discharged carbon support film (EMCN, Chicago, IL, USA) and stained with two droplets of 1% uranyl acetate for 5 and 25 s, respectively. The images were acquired with a JEM-2100 200 kV transmission electron microscope (JEOL, Akishima, Japan) equipped with an Orius camera (Gatan, Pleasanton, CA, USA).

## 3. Results

### 3.1. Sample Collection and Processing

At the Kunashir Island, samples were collected from the Valentina and Stolbovskiy springs located at the foot of the Mendeleev volcano. The Valentina springs represent a chain of small hot puddles and rills along the banks of the Valentina stream. The Stolbovskiy hot springs encompass two hot rills located close to each other and flowing into the Zmeiny stream [10]. Both Valentina and Stolbovskiy hot springs are located in a deciduous forest. Their waters are littered with fallen leaves and woods, which could serve as a substrate for heterotrophic thermophilic organisms. In Georgia, samples were collected from two hot springs that flow from artificial wells drilled in the 1970s during prospecting for coal deposits. The Tsaishi spring gives rise to a hot rill that flows through a grassy meadow into the Dzhumi river. The Nokalakevi spring includes two wells located next to each other and jointly supplying a rill flowing through a meadow into the Tekhuri river. This place is popular among local people and tourists and thus is a subject of a heavy anthropogenic load. In both Tsaishi and Nokalakevi, samples were collected from rills flowing on the ground. All samples represented mixtures of water and sediment (sand, soil, detritus). Water temperature and approximate pH were measured during sample collection and are listed in Appendix A. 

To enrich for bacteriophages, aliquots of environmental samples were added to liquid cultures of *T. thermophilus* strains HB8 and HB27. These two strains differ from each other by the set of defensive systems they harbor (Appendix A) [23]. After overnight cultivation at 70 °C, serial dilutions of culture supernatants were plated on lawns of each strain. Plaques of different sizes and/or opacity were purified and used to prepare phage lysates from which DNA was isolated (see Section 2). DNA from 10 isolates with distinct restriction digestion patterns was sequenced on the DNBSEQ-G400 platform with paired reads. The genomic assemblies yielded circular contigs suggesting that they represented complete viral genomes, either circular or with terminally redundant ends. Below, we present an overview of the genomes of the newly isolated viruses.

### 3.2. Filamentous Phages from Kunashir and South Caucasus

Two phages from the Tsaishi and Valentina hot springs produced tiny (less than 0.5 mm) turbid plaques and did not induce noticeable lysis upon inoculation of *T. thermophilus* cultures. Cultures infected with both phages contained thin ~800 nm filamentous particles with one round and another bud-like end (marked with black arrows on Figure 1A, where a representative image of culture infected with the Tsaishi isolate is presented). Similar bud-like structures were previously observed in *Escherichia coli* filamentous phages and considered to be adsorption complexes [24,25]. 

The DNA extracted from cell-free supernatants of infected cultures (and, therefore, containing virions) could not be sequenced with our standard “tagmentation” procedure designed for sequencing libraries preparation from double-stranded DNA. However, cells infected with both phages contained small DNA molecules that (i) were absent from uninfected cells and (ii) could be extracted from infected cells using plasmid purification kits. We, therefore, assume that the phages have single-stranded DNA in their virions and that extractable DNA from infected host cells represents replicative forms. 

Assembly of sequenced extracted circular DNA yielded circular contigs of 6773 (Tsaishi isolate) and 6680 (Valentina isolate) bp. We will refer to the corresponding phages as Zuza8 and Zuza27. The two phages demonstrate strict host specificity: Zuza8 produces plaques only on *T. thermophilus* HB8 lawns, while Zuza27—only on HB27. Alignments of phage-encoded proteins revealed that both phages are similar to each other and to previously published genomes of *T. thermophilus* phages φOH16 (Genbank ID: LC210520.1) and φOH3 (Genbank ID: NC_045425.1) isolated on Kyushu Island in Japan. The latter phage was studied in some detail and its virion components were annotated using proteometric approaches [26]. The genomes of all four phages share the same order of predicted open reading frames (ORFs). φOH3 is distinct from the other three phages due to the absence of an IS110-family transposase gene (Figure 1B). Since in their alignable parts the φOH3 and φOH16 genomes are highly similar at the nucleotide level (ANI~97%), and the Georgian phages, which are less similar to both each other and to the Japanese phages, contain the IS110-family transposase gene, φOH3 must be derived from a φOH16-like phage through transposase gene loss.

Integrated proviruses containing the entire set of homologs of Zuza8/Zuza27/φOH16 protein coding genes are present in the genomes of several *Thermus* isolates collected around the world (Honshu Island, Japan; Yellowstone Park, USA; Tibet; China; Kunashir Island, Russia), with the endonuclease gene located at one end and the IS110-family transposase gene—at the other end (Appendix A). The same organization was previously observed in proviruses of *Neisseria* filamentous phage Nf, which is thought to rely on its IS110-family transposase for integration and excision [27]. Thus, it is likely that Zuza8, Zuza27, and φOH16 also integrate into the genomes of their hosts in the same way. Lysogenization mediated by the IS110-family transposase, however, is not responsible for the apparently chronic character of host infection, since φOH3 forms turbid plaques [26]. Like with other inoviruses, the turbid plaques are likely produced by reducing the growth of infected cells. 

φOH3 infects the HB8 but not the HB27 strain [26]. The φOH3 and Zuza27 attachment proteins are homologous over their entire lengths. The N-terminal domain of the Zuza8 attachment protein could not be aligned by BLASTP to the corresponding parts of φOH3 and Zuza27 homologs (Figure 1B). However, Clustal Omega alignment revealed similarity in this region as well (Appendix A). The nature of host specificity of these viruses requires further investigation.

### 3.3. Lalka Phages—A New Group of Thermus Myoviruses

Samples from Valentina hot springs yielded three distinct lytic phages, Lalka8 (plaques on HB8 only) and Lalka27a/Lalka27b (plaque on HB27). Genomic DNA sequencing and assembly yielded circular contigs of 72,273 (Lalka8), 71,251 (Lalka27a), and 69,092 (Lalka27b) bp. Sequence analysis revealed a high level of similarity between the three genomes (ANI > 98%). Transmission electron microscopy with negative staining of lysates of *T. thermophilus* HB8 culture infected with Lalka8 revealed typical myovirus-like virions with isometric icosahedral heads (~80 nm in diameter) with thick ca. 100 nm tails. Most virions were associated with spherical objects (Figure 2A, left), which likely represent extracellular membrane vesicles released by *T. thermophilus* cells [28]. In some cases, rods coming out from thick sheaths of phage virions tails were visible (Figure 2A, right). 

The virion morphology suggests that Lalka8, as well as the closely related Lalka27a and Lalka27b have linear double stranded DNA genomes. The circular topology of assembled contigs, therefore, likely results from redundant genomic termini. A search for homologous nucleotide sequences in NCBI nr/nt database (July 2024) with blastn revealed no similarity to previously isolated *Thermus* viruses except for a segment encoding a methyltransferase of phage phiMa, previously isolated from El Tatio hot springs in Chile [4]. The corresponding proteins are 72% similar (DNMT in Appendix A). At the level of amino acid sequences, some Lalka phage proteins do show 30–65% identity to several other phiMa-encoded proteins (Appendix A).

Both Lalka8 and Lalka27a encode integrase genes followed by downstream sequences matching the 3′ end segments of *Thermus* alanine tRNA gene (Appendix A). In phiMa, the integrase gene is also present, followed by a sequence matching the 3′ end segment of *Thermus* isoleucine tRNA gene. Consistently, in *T. thermophilus* SNM6-6 from Senami (Honshu Island, Japan) and in a metagenome assembled genome (MAG) of *T. antranikianii* RBS10-92 from Kemerovo, Russia, very close relatives of phiMa are integrated at the isoleucine tRNA genes (Appendix A) [29,30]. It thus follows that Lalka8 and Lalka27a are capable of integration into the genomes of their hosts using the alanine tRNA gene as an attachment site. Use of different tRNA genes as attachment sites by related phages should allow to avoid interphage competition during lysogenization [31]. 

Upstream of Lalka8, Lalka27a, and phiMa integrase genes, there is a co-transcribed LexA-like transcription regulator gene and a gene encoding another transcriptional regulator that is transcribed in the opposite direction and that precedes a module or replication genes (Figure 2 and Appendix A, see also below). These divergent transcription regulator genes loci resemble the lysis–lysogeny switches observed in numerous phages including λ, P2, and 186 [32,33]. The presence of LexA-like transcription regulators suggests that the lysogeny-to-lysis transition in Lalka8 and Lalka27a could be induced by cellular response associated with DNA damage. Lalka27b lacks the integrase gene and the alanine tRNA gene attachment site but encodes the LexA-like protein and the transcription regulator (Appendix A). Thus, Lalka27b should be incapable of integration into the host genome. However, we cannot exclude a possibility that, similar to *E. coli* phage P1, Lalka27b may be maintained in lysogenic cells in a plasmid form [34].

In phage P1, virion-packed linear cyclically permuted DNA with redundant termini undergoes circularization with the assessment of tyrosine site-specific recombinase Cre [35]. All three Lalka phages, but not phiMa, encode tyrosine XerD-like recombinases (Figure 2 and Appendix A), which may perform this function. 

The Lalka genomes do not encode recognizable DNA polymerases, suggesting that they rely on host enzymes for replication. However, they encode several auxiliary replication proteins, namely, a RecT-like protein, a DNA polymerase processivity factor (β-clamp), a replication initiation O-like protein, a replicative helicase, and primase. Among these, only a homolog or RecT is present in the phiMa genome. Lalka replication-related genes, along with several genes with unidentified functions, cluster downstream of a transcriptional regulator gene and likely form a single operon. Outside this module, Lalka phages encode ssDNA-binding proteins which could also participate in phage DNA replication. The products of another Lalka phages module, an exonuclease, a nucleotidase, a dNMP kinase, and an RNA ligase, are probably involved in nucleic acids turnover (Figure 2). However, the RNA ligase genes have been partially lost from the Lalka27a/Lalka27b genomes and only short ORFs homologous to the C-terminal part of the Lalka8 RNA ligase remain (Appendix A).

All three Lalka phages encode large (2074 aa) proteins that contain methyltransferase and helicase domains and are similar with the DarB protein of phage P1. Similar to P1, the genes of DarB-like protein genes are followed by short genes encoding Ulx homologs (Figure 2 and Appendix A) [36]. The DarB and Ulx proteins are packaged in P1 virions and are required for defense against Type I restriction–modification systems of the host [36,37]. We hypothesize that DarB and Ulx homologs are also packaged in Lalka virions and overcome Type I restriction–modification systems that some *Thermus* isolates are known to harbor.

The three Lalka phages harbor short CRISPR-like arrays containing similar 29 bp repeats separated by 32–33 bp spacers. Since the consensus Lalka repeat sequence (GTTYYYCCCGTGCGCGCATGGAGGCACCG) does not match known *Thermus* CRISPR repeat sequences [4], it is unlikely that phages rely on host Cas proteins. Curiously, the putative Lalka CRISPR arrays are located next to TnpB-like transposase genes (Figure 2B and Appendix A). Some TnpB proteins are RNA-guided DNA nucleases [38,39]. Phylogenetic analysis suggests that TnpB proteins have been recruited as effectors of Type V CRISPR-Cas systems on numerous independent occasions [40]. The Lalka phage-encoded *tnpB*-CRISPR loci may thus be a novel thermophilic Type V CRISPR-Cas system. However, Lalka phages do not encode homologs of *cas1/cas2* genes which are essential for acquisition of new spacers.

Lalka8, Lalka27a, and Lalka27b encode DNA methyltransferases (DNMT), which, as mentioned above, are highly similar to the phiMa homolog. Lalka27b harbors an additional gene encoding a diverged (AAI 34% to Lalka DNMT) methyltransferase DNMTb. Lalka27a, in its turn, encodes DNMTa, which is barely related to either DNMTs or DNMTb. However, the DNMTa gene is highly similar (ANI > 90%) to an orphan DNA methyltransferase encoded in the genomes of many *T. thermophilus* isolates. In *E. coli* phage P1, viral DNA methylation mediated by either host-encoded Dam or phage-encoded Dmt methylase is essential for efficient production of infectious phage progeny [41]. Since the DNMT proteins are conserved among Lalka phages and a distant phiMa phage, they may be essential for the life cycle of these phages. In contrast, the functions of auxiliary methyltransferases DNMTa and DNMTb are likely nonessential. These enzymes may modify phage DNA to allow evasion from restriction–modification systems of specific host strains [42].

### 3.4. Cumys, a Phage from Kunashir Closely Related to the phiMa Phage from Chile

Another phage from Valentina hot springs, Cumys, was isolated on *T. thermophilus* HB8. It is very close (ANI 96%) to phage phiMa. Despite the high degree of overall similarity, there is a ~1900 bp region encoding a DarTG toxin-antitoxin system in phiMa that is absent in Cumys (Figure 3A). DarTG loci are also present in phiMa-like prophages found in the genomes of some *Thermus* isolates (Appendix A). In *E. coli*, DarTG protects cells from phage infection [43]. It is thus possible that *Thermus* cells lysogenized by phiMa-like phages are also protected from infection by other phages due to *darTG* loci.

Both Cumys and phiMa harbor a locus encoding a radical SAM (rSAM) protein and two proteins of unknown functions, of which one contains a DUF483 domain (Figure 3A, rSAM module). Similar loci are found in prophages of several bacterial genomes, implying a functional connection between the three genes (Figure 3B). According to the InterPro database [44], there exist fusions of DUF483 domains with methyltransferase or rSAM domains, which supports the functional association between the Cumys/phiMa rSAM module genes. Curiously, homologs of Cumys/phiMa rSAM and DUF483-domain proteins are also detected in several cyanobacterial (e.g., *Geitlerinema*) genomes in a context of genes involved in DNA modification with deazapurines [45] (Figure 3B). Although *Thermus* phages lack genes involved in deazapurine DNA modification, the rSAM locus may be involved in the introduction of other, yet unknown, DNA modifications.

### 3.5. Riverbug, a Phage from Kunashir, Closely Related to a Prophage of T. brockianus SNM4-1 Isolated from Japan

The Valentina hot springs yielded another phage, named Riverbug, which infects *T. thermophilus* HB8. This phage is very close (ANI 93%) to a prophage of *T. brockianus* SNM4-1 from the Senami hot spring on Honshu Island in Japan [30]. However, the prophage contains a ~2800 bp region that is absent from the Riverbug genome. This region harbors a transposase gene. Riverbug, in its turn, contains a ~1700 bp region that is absent from the *T. brockianus* SNM4-1 prophage and encodes two genes of unknown functions (Figure 4).

### 3.6. Three P23-45-like Bacteriophages from Stolbovskiy Hot Springs

Samples from Stolbovskiy hot springs yielded three closely related (ANI > 93%) phages, P23-45_stolb1, P23-45_stolb2, and P23-45_stolb3. These phages are similar (ANI > 85%) to a long-tailed phage P23-45 previously isolated from hot springs at the Valley of Geysers in Kamchatka peninsula [46]. The P23-45 genome is 86,364 bp long and contains two direct terminal repeats [5]; the non-redundant genomic sequence is 84,201 bp. The non-redundant genomes of P23-45_stolb1, P23-45_stolb2, and P23-45_stolb3 are shorter by ~8–10 kbps (76,044, 75,309, and 74,147 bp, respectively). Alignment between the four genomes revealed that the decreased size of Stolbovskiy phages is due to deletions in three regions of the P23-45 genome (Appendix A). In P23-45, these variable regions (VRs) contain 24 short genes, which encode proteins with unknown functions, except for host RNA polymerase inhibitor gp76 [47] located in VR1, and a cytosine methyltransferase in VR3. While gp76 homologs are conserved in all P23-45-like phages, the cytosine methyltransferase gene is absent in Stolbovskiy-derived phages, suggesting that this protein is not essential. P23-45 and P23-45-like phages from Stolbovskiy hot springs encode conserved ribonucleotide reductase proteins. In P23-45_stolb1, the ribonucleotide reductase contains a 230 aa insertion representing an intein-containing homing endonuclease.

### 3.7. A South Caucasus Phage Related to Phage IN93 from Japan

A phage infecting *T. thermophilus* HB8 was isolated from the Nokalakevi hot springs located ca. 30 km from Tsaishi. This phage, Tsatsa, produced large, 3–4 mm, plaques and rapidly lysed liquid cell cultures. Sequencing and assembly yielded a circular 20,815 bp contig. Extensive similarities of Tsatsa proteins to those encoded by a previously described *T. thermophilus* phage IN93 (Genbank ID: NC_004462.2) isolated on Honshu Island in Japan [48], were observed. Genes encoding homologous proteins are colinearly located in the genomes of the two phages (Figure 5). Like Tsatsa, IN93 infects the HB8 but not the HB27 *T. thermophilus* strain [49].

IN93 is a lysogenic phage; it integrates into the host genome through an attachment site located in the Ile-tRNA gene [49]. A sequence matching 45 bp of the 3′ terminal part of *T. thermophilus* Ile-tRNA is present in the Tsatsa genome immediately downstream of the integrase gene, implying that both phages integrate into the same site. Interestingly, though Tsatsa and IN93 are unrelated to phiMa-like phages, their integration sites are identical. Yet, the identity between the corresponding integrases is only ~40%. 

Although Tsatsa and IN93 genomes share a collinear order of predicted protein coding genes, several differences exist. For example, in the Tsatsa genome, there is an insert between the homologs of adjacent IN93 genes *2* and *3* located close to the left end of the genome (Figure 5). The insert contains a gene encoding a protein similar to phosphoadenosine phosphosulfate (PAPS) reductases, enzymes involved in sulfur metabolism. PAPS reductase homologs were observed in various mobile genetic elements, including bacteriophage λ [50], marine bacteriophages [51], and archaeal viruses and plasmids [52,53,54]. A gene encoding a PAPS reductase homolog is located, in the same position as in the Tsatsa genome, in the genome of *Thermus* phage P23-77, a more distant relative of IN93 [55]. PAPS reductase homologs are also associated with Type IV-B CRISPR-*cas* [56] Type IV BREX [57], and phosphorothioation-based prokaryotic defense systems. In the latter, PAPS reductases are presumably involved in the DNA modification process [58]. Since PAPS reductases are homologous to diverse ATP pyrophosphatases, their functions may extend beyond sulfur metabolism [59]. Clearly, these interesting phage proteins deserve further investigation.

Comparison between the Tsatsa and IN93 genomes revealed a variable region located between genes encoding an integrase and a LexA-like protein (Figure 5). In IN93, this region contains two genes. The product of the first gene has no detectable homology with characterized proteins but contains a predicted N-terminal lipoprotein signal peptide. The second encodes a protein with an N-terminal domain resembling Type IV restriction endonucleases followed by a membrane-spanning segment and a predicted extracellular DUF4101 domain of unknown function. The corresponding region in Tsatsa contains four genes (Figure 6). The product of the first gene resembles periplasmic TolB proteins and contains six WD40 repeats and a predicted N-terminal signal peptide [60]. The fourth gene encodes a small (126 aa) HEPN-domain protein. 

HEPN-domain nucleases are known to participate in numerous defense systems [61], while WD40 repeat proteins are involved in protein–protein interactions and signal transduction [62]. We propose that in both phages, this variable genomic region encodes defense systems. This prediction is almost certainly valid for IN93, given that Type IV restriction endonucleases are known to degrade modified DNA molecules and thus the enzyme may protect IN93 lysogens from bacteriophages with modified DNA [63]. To perform this function, the IN93 Type IV restriction endonuclease gene has to be expressed in the lysogenic state. Indeed, genes encoding the LexA-like regulator, the putative periplasmic protein, the restriction endonuclease, and the integrase comprise an operon and are thought to be transcribed in the IN93 prophage [49].

Inspection of IN93-like proviruses retrieved from sequenced *Thermus* genomes revealed in many the presence of genes encoding restriction endonucleases, HEPN domain proteins similar to that of Tsatsa, as well as variable sets of neighboring genes encoding putative lipoproteins and proteins with predicted extracytoplasmic localization or transmembrane proteins containing predicted extracytoplasmic domains (Figure 6). We propose that the HEPN-domain nucleases are effectors of putative phage defense systems active in the lysogenic state. While the functions of other proteins are unknown, we hypothesize that they are involved in superinfection exclusion as some superinfection exclusion proteins were shown to be lipoproteins [64] or transmembrane proteins [65] expressed in the lysogenic stage. Indeed, in several phage genomes, genes encoding superinfection exclusion proteins are flanked with LexA-like repressor and integrase-encoding genes [66].

Interestingly, in a completely unrelated bacteriophage Riverbug, as well as the Riverbug-like prophage from *T. brockianus* SNM4-1, a gene encoding a protein homologous to Type IIL restriction-and-modification enzymes and another encoding a protein with predicted transmembrane and extracytoplasmic domains is also located between a LexA-like repressor and integrase genes (Figure 6). It thus follows that such modules are capable of horizontal transfer between unrelated phages.

## 4. Conclusions

The impetus for this study came from a hypothesis that the variety of *Thermus* phages is undersampled. Indeed, the 10 *Thermus* phages presented here show remarkable diversity and include both novel (Lalka phages) and very close relatives of known viruses previously isolated from around the world. Phages from the latter group are globally distributed and it is interesting to determine how they spread given the highly fragmented nature of *Thermus* habitat. Even closely related viruses contain variable regions that harbor known or hypothetical defense systems. Functional analysis of these systems, including the putative novel CRISPR-Cas of Lalka phages, may provide valuable insights for both basic and applied science. 

## Figures and Tables

**Figure 1 viruses-16-01410-f001:**
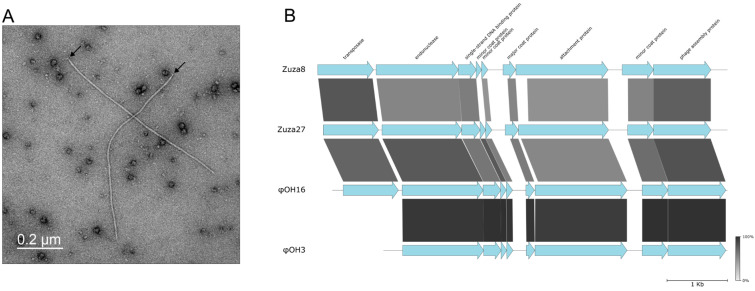
Filamentous phages from Tsaishi and Valentina hot springs. (**A**) Zuza27 virions in infected *T. thermophilus* HB27 cultures visualized with transmission electron microscopy with negative staining. (**B**) Graphical alignment of the genomes of Zuza8 (from Tsaishi hot spring in Georgia), Zuza27 (from Valentina hot spring in Kunashir), and φOH16/φOH3 phages from Kyushu Island in Japan. ORFs are represented with arrows that indicate the direction of transcription. Zuza8 and Zuza27 ORFs whose products’ functions could be predicted by homology are colored blue; the predicted functions are listed at the top. The homologous gene segments shared between the viruses are connected with bars of different shades of gray based on the amino acid sequence identity levels shown in the bar at bottom right.

**Figure 2 viruses-16-01410-f002:**
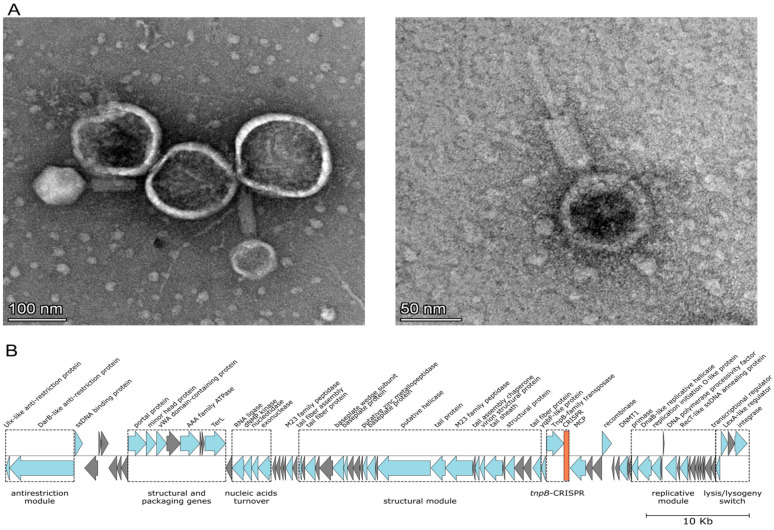
The Lalka8 phage from the Valentina hot spring in Kunashir. (**A**) Lalka8 virions visualized with transmission electron microscopy with negative staining. (**B**) A graphical map of the Lalka8 genome. See Figure 1B legend for details. Groups of genes encoding proteins of similar functions form modules indicated with dotted line rectangles; a putative CRISPR array is shown in red.

**Figure 3 viruses-16-01410-f003:**
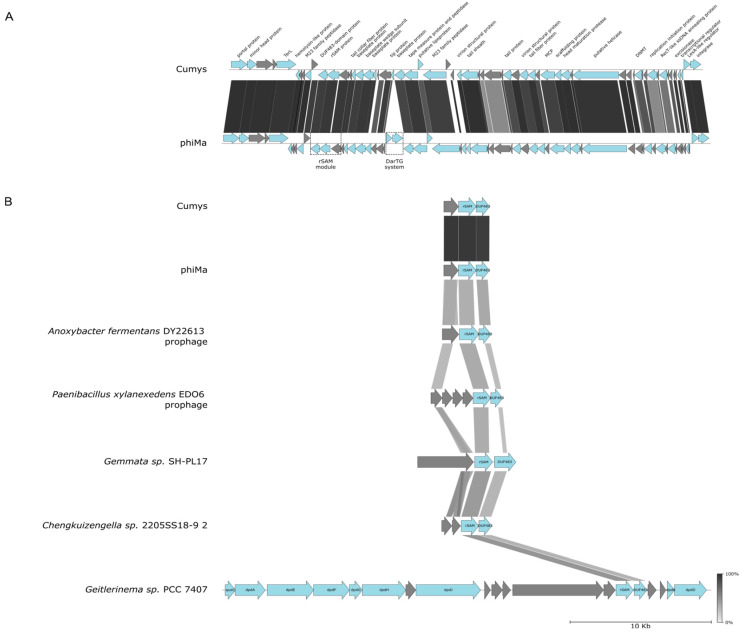
Phage Cumys from Valentina hot springs. (**A**) A graphical map of the Cumys genome. See Figure 1B legend for details. (**B**) Loci similar to Cumys radical SAM (rSAM) protein gene module are present in diverse phages and prophages.

**Figure 4 viruses-16-01410-f004:**
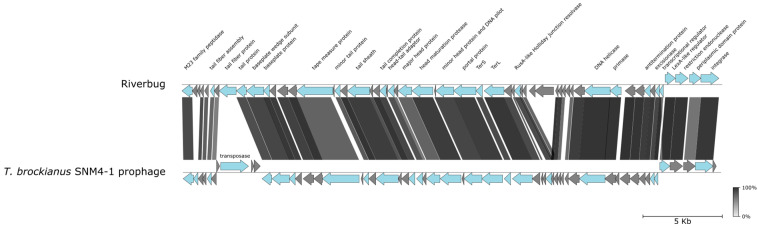
Phage Riverbug from Valentina hot springs. Graphical alignment of the Riverbug genome with the *T. brockianus* SNM4-1 prophage. See Figure 1B legend for details.

**Figure 5 viruses-16-01410-f005:**
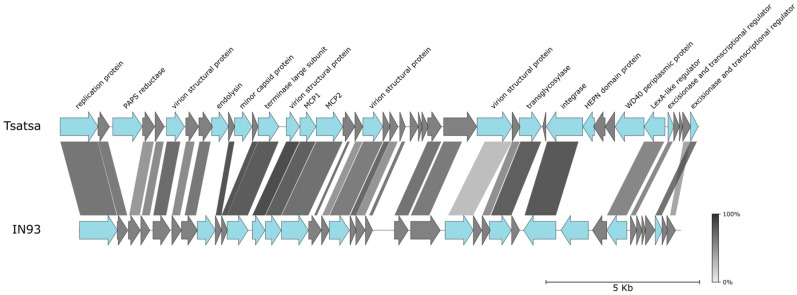
Phage Tsatsa from the Nokalakevi hot spring (Georgia). Graphical alignment of Tsatsa and IN93 genomes. See Figure 1B legend for details.

**Figure 6 viruses-16-01410-f006:**
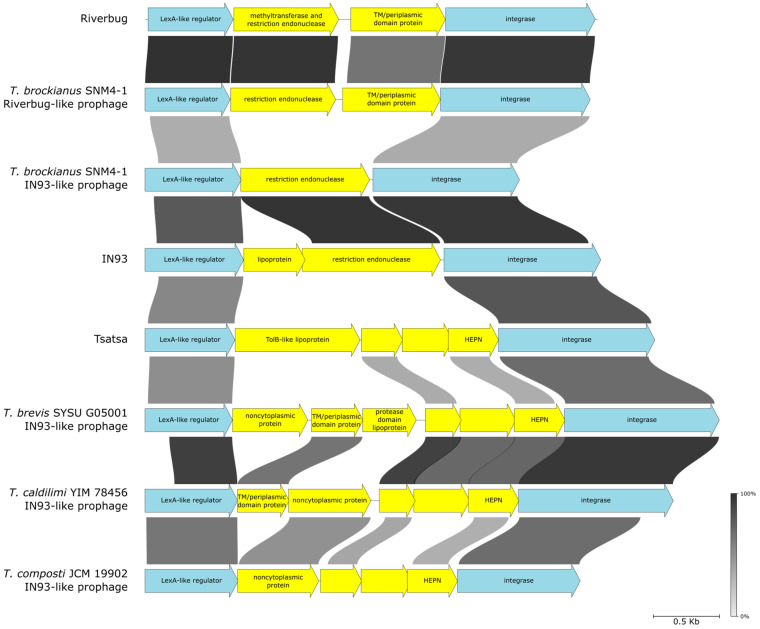
Putative phage-encoded defense systems located in regions likely expressed in a lysogenic state. Graphical alignment of diverse phage genome segments flanked by genes encoding a LexA-like regulator and an integrase. Genes constituting putative defense systems are colored yellow.

## Data Availability

Genomic sequences of bacteriophages Zuza8, Zuza27, Lalka8, Lalka27a, Lalka27b, Tsatsa, Cumys, Riverbug, P23-45_stolb1, P23-45_stolb2, and P23-45_stolb3 are deposited in NCBI GenBank under the accession numbers PP554188, PQ109074, PQ109075, PQ109076, PQ109077, PQ109078, PQ109079, PQ109080, PQ158664, PQ158665, and PQ158666.

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
