# Peer review of "New Viruses Infecting Hyperthermophilic Bacterium Thermus thermophilus"

_viruses, 2024, doi:10.3390/v16091410_

Round 1

Reviewer 1 Report

Comments and Suggestions for Authors

The authors isolate and sequence several novel phages from Thermus thermophilus bacteria and compare them to the known diversity of phages from this genus.

The authors should try to taxonomically place their phages using a program such as https://github.com/amillard/tax_myPHAGE or ClassiPhage (https://www.mdpi.com/1999-4915/11/2/195). What I would appreciate is a table of all isolated phages, host, sample source and taxonomy.

Other than that, the work is solid, represents a welcome addition to our knowledge of phages, and I have only minor comments for fixes before publication.

Line-by-line:

29 (and several other instance) – double spaces should be replaced by single spaces (nitpick)

39: I don’t doubt that Thermus phages are undersamples (most phages are), but an abundance of CRISPR-spacers that don’t match known phages is not evidence  for that – it could be plasmids or other mobile genetic elements.

97: What’s the MGI platform?

110: Could they only be extracted with plasmid miniprep kits? It should be noted and not just implied that these phages have ssDNA genomes in their virions, but (extractable) dsDNA replicative forms in host cells.

135: It might be worth noting here that inoviruses do not lyse their hosts but produce “plaques” for (pseudo)lysogens by reducing the growth of infected hosts.

136-139: A supplementary figure showing the alignments of the attachment proteins would be appreciated here

174: Do the authors actually mean similarity or identity? These terms have different meanings, and I suspect they mean identity.

304: Nitpick, but IN93 is not technically a sphaerolipovirus anymore as far as I can tell, in the sense that they are not in the Sphaerolipoviridae family – It’s a Matsushitoviridae (who can really keep track of all these changes?)

352: It is never described how prophages were detected in Thermus genomes – add this to the methods

364: Again what do the authors mean with similarity? Homology? Similar annotation? I do not see evidence in figure 6 or the text that these restriction endonucleases are shared between unrelated phages

395: Which Thermus strains? They’re mentioned in the text, but should also be here

Reviewer 2 Report

Comments and Suggestions for Authors

This is a well-written paper that significantly expands our knowledge of the occurrence, morphology and genetics of phages infecting the Thermus genus. It is rare that I read a paper without suggesting improvements.
